# Indonesian Scientists' Behavior Relative to Research Data Governance in Preventing WMD-Applicable Technology Transfer

Lindung Parningotan Manik [1,*], Zaenal Akbar [2], Aris Yaman [2] and Ariani Indrawati [2]

[1] Research Center for Data and Information Sciences, National Research and Innovation Agency, Bandung 40135, Indonesia
[2] Research Center for Computing, National Research and Innovation Agency, Bogor 16911, Indonesia
[*] Correspondence: lind008@brin.go.id

**Abstract:** Performing research data governance is critical for preventing the transfer of technologies related to weapons of mass destruction (WMD). While research data governance is common in developed countries, it is still often considered less necessary by research organizations in developing countries such as Indonesia. An investigation of research data governance behavior for Indonesian scientists was conducted in this study. The theories of planned behavior (TPB) and protection motivation (PMT) were used to explain the relationships between different factors influencing scientists' behavior. The theories have been widely used in the information security domain, and the approach was adopted to build the research model of this study. The obtained data were analyzed using partial least-squares structural equation modeling (PLS-SEM) to answer the main research question: "what factors determine the likelihood of practicing research data governance by Indonesian scientists to prevent WMD-applicable technology transfer?" By learning what motivates scientists to adopt research data governance practices, organizations can design relevant strategies that are directed explicitly at stimulating positive responses. The results of this study can also be applied in other developing countries that have similar situations, such as Indonesia.

**Keywords:** research data; data governance; weapon mass destruction; technology transfer; theory of planned behavior; protection motivation theory





## 1. Introduction

Data have become an integral part of our daily activities in today's digital era. We continuously produce and consume data to support our decision-making processes. As an asset, the increase in volume, the variety of formats, and the veracity of data quality require a systematic solution for handling data correctly. A solution for that is via data governance, a process of applying authority and control over data such that the data can function adequately and ensure accountability [1]. Data stakeholders are individuals, groups of individuals, or organizations affected by how data are governed [2]. They hold the privilege of knowing how data are collected and treated, and the values, interests, and norms regarding its use. Stakeholders gain different benefits from the value of the type of data, such as being able to perform data aggregation and analytics [3]. The primary purpose of data governance is to increase the value of the data and at the same time reduce data-related costs and risks [4]. Within organizations, data governance is highly sensitive to the domain and actors involved in the organization's activities. As an asset to an organization, data-driven businesses are highly affected by three key business elements: use, design and storage, and processes and people [5]. In this case, data governance refers to the assignment of control for different decision domains where models, policies, and standards governing how data are stored, integrated, and put to use are dictated by the element's processes and people.

On the other hand, Indonesia's government and scientific community are increasingly eager to share and provide access to their digital data. The government established the One Data Indonesia[1] (ODI) portal to realize an open government data initiative since 2016 [6]. Moreover, the Indonesian government research agency, the National Research and Innovation Agency (BRIN), has built the National Scientific Repository[2] (RIN), enabling Indonesian scientists to share their research data with the public [7]. There are 99,138 datasets shared in ODI and 4659 datasets shared in RIN (accessed on 21 September 2022).

The ODI's portal is an implementation of the Indonesian presidential decree number 39[3] (from 2019), which is related to the government's data policy, and the purpose of the portal is to create quality data that are accessible and shared across organizations. Previously, the issue of research data governance in Indonesia was addressed by the code of ethics for research activities regulated in the Ministry of Research and Education's decree of 25/M/Kp/III/2013[4] from 2013. Moreover, Indonesian law number 11[5] (from 2019), with respect to the national science and technology system, requires that any primary data, including the output of research activities, be stored in an integrated information system. Furthermore, the regulation of BRIN number 18[6] (from 2022) affirms RIN as an integral part of the national science and technology information system. Indonesian law number 11 of 2019, Indonesian presidential decree number 39 of 2019, and BRIN regulation number 18 of 2022 express the need for standardized metadata, which is domain-dependent on enabling data reusability and interoperability.

Besides promoting transparency and accountability, the open data initiative is an excellent movement for big data analysis and public benefit [8]. However, it can also be a double-edged sword that is simultaneously capable of significantly advancing science and technology and causing backlash if not appropriately utilized [9], which we call the dual use of research concern (DURC). Weapons of mass destruction (WMD) applications are research data misuses that should be prevented [10]. WMD-applicable research concerns scientists from various disciplines, such as biology, biotechnology, chemical, nuclear science, artificial intelligence, and advanced computing (among others). Scientific communities have significant roles in this dual use case. For example, the biomedical community is concerned with the threat of biological warfare and terrorism. At the same time, the genomics revolution holds great promise for advancing basic biology, medicine, and agriculture [11]. While the same advances in microbial genomics could be used to produce bioweapons, they can also be used to set up countermeasures against them. Therefore, the scientific community plays a crucial part in generating a network of deterrence. Furthermore, identifying community roles would be essential for a data governance strategy.

The sources or materials produced, generated, and compiled in research are referred to as research data [12]. Data are used to provide answers to particular research questions. It could be in the form of clinical records, genetic sequences, specimens, samples, videos, audios, texts, images, documents, spreadsheets, and so on. Since research data are strategic assets, they must be handled well. Creating and applying rules to preserve and maximize data value are examples of research data governance practices [13]. Unwanted risks can arise as a result of a lack of data governance.

In a study conducted in the United States (US), 65% of respondents believed that individual scientists should be primarily responsible for the research data [14]. Thus, awareness and continuous education or training should be raised within scientific communities to protect the research data against theft, forced transfer, or predatory acquisition. Besides the scientists, academic institutions also have significant primary responsibilities. Therefore, policies and strategies are needed within the organizations to guide their staff in managing research data [15].

In this study, we investigated Indonesian scientists' research data governance behavior in preventing the transfer of WMD-applicable technologies based on their awareness of WMD, their organization's policy on data governance, and their experience and involvement in governing research data. This study focused more on digital research data, and the discrete and discontinuous representations of information. This study was quantitative

research that applied the theory of planned behavior (TPB) and the protection motivation theory (PMT) as the backbone of the research model. The results were expected to give the governments, experts, and research organizations guidelines to improve their data governance and create strategies for promoting the research data governance behavior for the scientists. The results of this study could also be applied in other developing countries with similar situations, such as Indonesia. Developing countries encounter multiple challenges in implementing data governance principles mainly due to resource limitations and financial and skill limitations. In addition, the situation requires the merging of multiple organizations and firm collaboration, which introduces a complex dynamic environment within different countries [16]. Higher local and global data regulations, including policy discrepancies among organizations, are another challenge [17]. Above all, the lack of a national-level strategy impedes countries from proper and effective implementations [18].

As the object of this study, the implementation of data governance, especially for research data in Indonesia, is no exception. Currently, there are two governmental organizations for supervising research activities in Indonesia. First, research activities are performed by multiple non-ministerial government agencies, such as the Indonesian Institute of Sciences (LIPI) and the Agency for the Assessment and Application of Technology (BPPT). In 2021, those non-ministerial government agencies merged and integrated into BRIN. Second, Higher Education Institutions (HEIs) performed research activities as part of their three pillars of higher education (Tri Dharma Perguruan Tinggi), namely, education, research, and community services. In this case, the activities are supervised by the Ministry of Education and Culture. The quality of HEIs, including state and private universities, is assessed by the National Accreditation Body for Higher Education (BAN-PT). Accreditation is given to both program study and university level studies for five years. The issued accreditation levels are A (excellent), B (very Good), and C (good). For a national-level strategy in data governance, ODI portal and RIN enabled data sharing and interoperability among organizations.

Besides the main subject of this research study being novel, this study also addressed a gap in research data governance studies that seems limited. In developed countries, research data governance has become standard practice to ensure organizations' proper management of data assets. Data governance information can be viewed directly on the institutions' websites. Meanwhile, in developing countries such as Indonesia, the studies of research data governance are very limited. It remains unknown which research organizations have implemented research data governance practices. It also could be that there never was a formal and established policy in research organizations, since it is often considered unnecessary. However, scientists might have been practicing data governance on a daily basis. Research data governance behavior must be encouraged to prevent the dual use of research, and more specifically, to prevent the transfer of WMD-applicable technologies.

*1.1. Research Questions*

This research aimed to explore Indonesian scientists' research data governance behavior. The further aim of this research study was to study the main determinants of research data governance intentions in Indonesia. Therefore, the broad research questions are as follows.

**RQ 1.** *Do Indonesian scientists' practices in producing, storing, accessing, or sharing digital materials influence the necessity to practice research data governance?*

RQ 1 is concerned with understanding how Indonesian scientists' behaviors in producing, storing, accessing, processing, or sharing digital materials affect the need to practice research data governance. As objects that can be produced, stored, accessed, processed, or shared by computers, digital materials are double-edged swords. On the one hand, sensitive data should be protected and managed correctly, especially when modern tech-

nologies are used to handle data, such as cloud data storage [19]. On the other hand, data-sharing practices have advanced research and opened up numerous potential applications in various domains. However, the practices should also maintain data privacy and security, which have become major concerns. For example, in genomic data sharing [20] or disease-surveillance-data sharing [21], potential risks must be assessed as part of data governance and regulatory frameworks. One should guarantee that privacy and security are appropriately supported, especially in individual or population-level datasets. More than that, using the public cloud to process such data requires more protection, including privacy protection and data security [22].

**RQ 2.** *To what extent do Indonesian scientists' data handling practices meet the standards of good data governance practices?*

RQ 2 is concerned with understanding how Indonesian scientists perceive the numerous methods of handling digital materials in practicing data governance. Collecting and storing extensive amounts of data are parts of scientists' activities. However, it does not mean they are well-equipped to handle the data correctly, especially within public sector agencies [23]. Consequently, the confidence in the data can be reduced significantly, affecting the quality of service or products produced. Furthermore, the capability to govern data across organizations is challenging due to the heterogeneity of resources as a result of different actors [24]. As individuals, a group, or an organization, scientists who have collected and stored data are affected by how the data are governed and the value created from these data. Therefore, there is an urgent need to ensure that a data governance model is suitable for the situation. For example, the data-sharing pool model, a new emerging data governance model, dictates a key mechanism that defines data-sharing modalities, how data can be handled, and for what purposes they are used [2].

**RQ 3.** *Do practices of research data governance differ between institution types and institutions of different accreditation?*

RQ 3 is concerned with understanding how Indonesian scientists perceive the contribution of their institutional accreditation when practicing data governance. As a lasting set of assumptions, beliefs, and values that describe organizations and their members, organizational culture may positively influence information governance's effectiveness [25]. The culture, which can be profiled as collaboration, creation/innovation, controlling/hierarchy, and competition/result-oriented, requires trust as the enabler and driver of governance processes in an organization. In this sense, an organization's accreditation reflects the assurance of the quality of services provided by the organization, and accreditation contributes more to the improvement of processes and practices in institutions [26].

**RQ 4.** *What factors determine the likelihood of practicing research data governance by Indonesian scientists in preventing WMD-applicable technology transfer?*

RQ 4 is concerned with understanding factors that encourage or discourage research data governance intentions. In short, the main objective of this study was to examine Indonesian scientists' research data governance behavior. By learning what motivates scientists to adopt this form of behavior, organizations or experts could design relevant strategies directed explicitly at stimulating positive responses. As the implication for research and researchers, to the best of our knowledge, this study is the first investigation targeted at theory-driven scientists' research data governance behavior, especially in developing countries such as Indonesia. In general, theory-driven research promotes a greater understanding of the attitudes and behaviors that affect a particular action and eventually promotes the successful design and execution of interventions that attempt to promote the behavior. Thus, this study aimed to apply TPB [27] and PMT [28] in studying Indonesian scientists' research data governance practices.

*1.2. Literature Review*

A conceptual framework for data governance that covers antecedents, scoping parameters, and governance mechanisms that practitioners can use in approaching data governance was introduced by [4] in a structured manner. The framework covers multiple entities, including the scope of data, the scope of organizations, and the scope of the domain in which the data governance model will be applied. Data quality, security, architecture, lifecycle, metadata, storage, and infrastructure are examples of the domain scopes of data governance where most strategies address two or more of those areas. More than that, a study conducted by [2] examined four models for data governance, namely, data-sharing pools, data cooperatives, public data trusts, and personal data sovereignty.

The work in [4] dictates that, in order to promote desirable behavior in the use of data, a data governance strategy should develop data policies, standards, and procedures. Organizations tend to behave differently when optimizing the manageability of sustainability data [29]. A preliminary understanding of why regulating collective behavior related to data in an organizational context is also complex [24]. Researchers also examined the most up-to-date policy regime for handling data-driven systems, which is the new GDPR in the EU, to evaluate the extent to which the rule might justify ideal behavior and improve responsible data governance practices [30]. Other researchers examined human data capability chains, which proved the necessity and value of big data governance with respect to healthcare [18].

In many countries around the world, data governance has become a major issue with respect to the national security of a country. As an example, personal information protection and data disclosure/accountability strategies are required in order to provide accountable pension services in South Korea [31]. Moreover, implementing strong data governance structures and ensuring the ethical use and reuse of individuals' data collected via digital health programs are necessary, especially in low-income and middle-income countries [32]. Proper data governance could underpin urban smart cities and sustainable development solutions in several cities in European countries [33].

On the other hand, multiple organizational roles in data governance strategies have also been discussed. For example, the role of the board of directors within organizations has been extended to ensure that data are actively managed in an increasingly technology-intense environment [34]. At least four data management activities are required [35] to prevent data misuse. First, data security comprises the need to ensure that data, which are primarily confidential or sensitive, are stored securely with appropriate authentication and authorization mechanisms. Second, data preservation requires long-term data archiving with associated collection protocols. Third, data compliance is a requirement to adhere to the standards and policies of other relevant agencies, in addition to legal obligations such as data protection. Lastly, data sharing requires mechanisms and systems that allow open access to data where appropriate.

This work should provide a deeper understanding of the multitude of actions from stakeholders that will affect the practiced data governance strategies. Furthermore, it will also shed light on the quality of service provided, as reflected in how accreditation has contributed to the practice of data governance.

## 2. Theoretical Model and Hypotheses

This section describes the theoretical concepts adopted to formulate the research model and hypotheses for answering the main research question, RQ 4.

*2.1. Planned Behavior and Protection Motivation Theories*

TPB was developed to improve an earlier paradigm, namely, the theory of reasoned action (TRA) [36], to predict human behavior [37]. The models assume that individuals make rational, reasoned choices to engage in specific behaviors based on their available information [38]. In the past, TPB has been widely used in various settings, such as health-related issues [39], leisure intentions [40], and other domains. Currently, the theory is

also applicable to digital technology domains such as internet purchasing [41], mobile learning [42], and the intention to use social networking websites [43].

Every action an individual takes is motivated by three considerations according to the TPB [27]. First, behavioral beliefs (beliefs about the likely effects of the performed behavior) usually manifest positively or negatively toward a particular behavior. Second, normative beliefs (beliefs about the normative perceptions of other people) result in subjective norms. Third, control beliefs (beliefs about the presence of factors that may enable the performance of the behavior) trigger perceived behavioral control. Generally, the higher the attitude, subjective norm, and perceived behavioral control, the greater the individual's intention to execute the behavior.

The TPB is often insufficient for explaining all variances in intention and adherence behaviors [44]. Thus, in this study, we combined one more theory to predict data governance behavior. PMT is one paradigm that explores why individuals engage in risky habits and suggests ways to change those behaviors [45]. Initially, it reduces the likelihood of contracting health issues [46]. However, it is also applicable to climate change threats [47], waste management issues [48], etc.

Similarly to TPB, PMT also has been adapted in technology-related domains, especially with respect to security issues such as desktop security [49], backing up personal data [50], online safety behaviors [51], and others. The PMT proposes that individuals exhibit protection behavior based on two types of considerations: threat appraisal and coping appraisal [28]. Threat appraisal evaluates the seriousness of a threat, the possibility of being harmed by the threat (perceived vulnerability), and the determination of how dangerous it is (perceived severity). Meanwhile, a coping appraisal strategy assesses how the individual reacts to the situation. It consists of response efficacy, the effectiveness of the recommended behavior in eliminating or avoiding potential harm, and self-efficacy, which is the belief that an individual can effectively carry out the recommended behavior. Generally, the higher the threat and coping appraisal, the greater the individual's intention to execute the behavior.

The TPB and PMT are applicable for predicting information security behavior. For example, in [52], the TPB with anticipated regret and threat appraisal is sufficient for predicting the information security policy compliance of employees in a research organization. Furthermore, TPB and PMT were integrated to explain the compliance of business managers and information system professionals with respect to the information system's security policy (ISSP) [53]. The results indicated that the perceived severity and response costs were not determinants of ISSP behavioral compliance intentions. Meanwhile, in [54], all components of PMT and the habit (a routinized form of past behavior) significantly impacted employee intentions to comply with ISSP. Besides the compliance intention, the TPB has also been used in other information security aspects, such as the knowledge sharing model in organizations [55].

### 2.2. Research Model and Hypothesis

This study investigated Indonesian scientists' behavior in practicing research data governance in preventing WMD-applicable technology transfer based on TPB and PMT. The research model is presented in Figure 1.

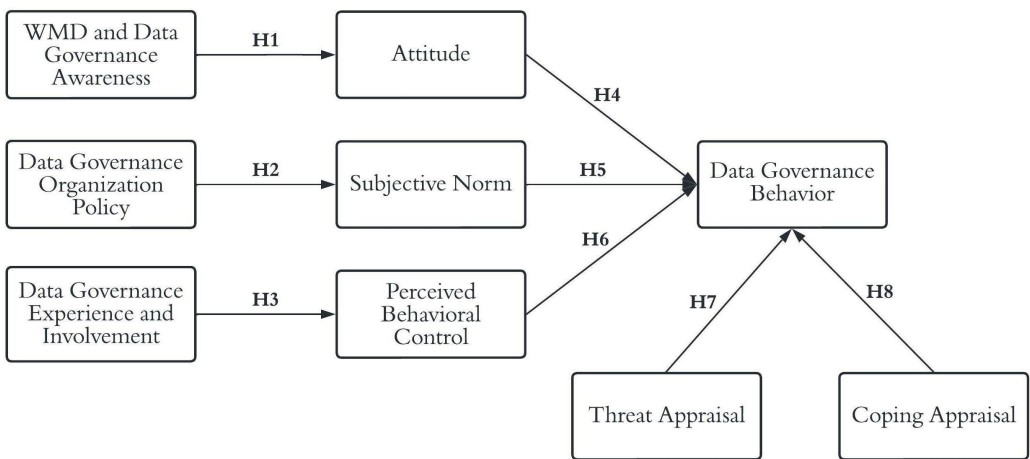

**Figure 1.** Research model.

An awareness-level understanding of WMD and research data governance's best practices, policies, and procedures is essential for scientists in governing research data in order to prevent the transfer of technologies with WMD applications. Furthermore, appropriate information about WMD is needed to raise user awareness, which contributes to safe behavior [56,57]. Experts can develop users' knowledge about the dual use of research via training classes, seminars, and organized lectures. We hypothesized that the awareness of WMD and best practices, policies, and procedures in research data governance positively influences attitudes toward performing research data governance activities.

**H 1.** *The awareness of weapons of mass destruction and best practices, policies, and procedures in research data governance positively affects attitudes toward performing research data governance activities.*

In sharing and preserving research data, data governance is essential in most research organizations. There are some tools available for performing data governance. Dataverse, DSpace, CKAN, and Zenodo are several tools used by organizations in governing research data. However, technological tools alone are insufficient. Scientists should also care about how they manage research data. To govern the data, organizations need to create policies and standards for operating procedures because information, specifically, WMD-applicable data, is considered an important asset that scientists should safeguard [14]. Scientists should be able to understand and follow data governance policies. Promoting positive behavior and discouraging poor user behavior can be effective organizational strategies [58]. Subjective norms reflect the influence of the view of significant others on individuals' decisions. This pressure affects users' behavior in organizations. We hypothesized that organizational policies positively affect subjective norms toward governing research data.

**H 2.** *Organizational policies positively affect subjective norms toward performing research data governance activities.*

Data governance involvement comprises scientists' time, energy, and effort invested in ensuring a secure data environment. Several studies investigated individuals' involvement in organizational activities and policies in the information security domain. According to the findings, information security engagement positively impacts users' attitudes toward information security policy compliance [53,59]. We also adopted this concept in the data governance domain. Furthermore, the involvement and the experience affect perceived behavioral control, which is the perceived ease or difficulty in performing certain behaviors [60]. The more involved and experienced the individual, the easier it is for scientists to govern the research data. We hypothesized that scientists' experience and involvement pos-

itively affect the perceived control of behavior toward performing research data governance activities.

**H 3.** *Scientists' experience and involvement positively affect perceived behavioral control toward research data governance activities.*

Based on the factors above, we hypothesized that three native TPB variables, such as attitude, subjective norms, and perceived behavioral control, positively affect the performance of research data governance activities. Attitude, the prominent item in TPB, is a critical aspect that affects an individual's behavior. In this context, it is a positive or negative expression toward performing research data governance activities. Subjective norms are the perceived social pressures to practice (or not perform) behaviors. They represent the influences of significant others' views on an individual's decision. Meanwhile, as explained before, perceived behavioral control is the perception of ease or difficulty in performing the behavior. If individuals evaluate research data governance behavior positively (attitude) and believe that other significant individuals expect them to execute that behavior (subjective norms) and also believe that it is easy to perform (perceived behavioral control), this leads to motivation. Therefore, they are more likely to exhibit that particular behavior.

**H 4.** *Attitudes toward research data governance positively affect performing research data governance activities.*

**H 5.** *Subjective norms have a positive effect on performing research data governance activities.*

**H 6.** *The perceived behavioral control has a positive effect on performing research data governance activities.*

Furthermore, we hypothesized that two native PMT variables, threat appraisal and coping appraisal, positively affect performing research data governance activities. These factors have been widely used in the information security domain, and we adopted them in this study. Threat appraisal is the perception of the possibility and severity of danger. Suppose individuals believe that not performing research data governance activities increases the probability of threatening events, and they believe that the consequences of the events endanger them. In that case, this leads to motivation. Meanwhile, the coping appraisal is the individuals' evaluation of their abilities in dealing with and avoiding harm due to the threat. Suppose individuals have the skills or measures needed to perform research data governance activities. They believe they can avoid WMD-applicable transfer technologies by performing that particular behavior. In that case, this also leads to motivation.

**H 7.** *Threat appraisal has a positive effect on performing research data governance activities.*

**H 8.** *Coping appraisal has a positive effect on performing research data governance activities.*

## 3. Materials and Methods

This section explains the population, sampling strategy, data collection methods, research instruments (questionnaires), and analysis methods.

### 3.1. Data Collection

Data were collected using online questionnaires[7]. The invitations to fill the questionnaires were sent by email to the selected participants chosen from a database of Indonesian scientists, which are stored in the SINTA's[8] (Science and Technology Index) information system [61]. We sent a link and an access code for participants to fill out the questionnaires in the email. We also attached informed consent, an approval letter of ethical clearance

from the ethics committee at our institution, and a cover letter from our research center's director, which explained our study's urgency. Moreover, the reminder emails were sent three times.

Most Indonesian scientists, about 97% of the population, are based in higher education institutions (HEIs). The Indonesian Ministry of Education sets each HEI an accreditation, such as "A" or "excellent," "B" or "very good," and "C" or "good" [62]. Meanwhile, small Indonesian scientists are affiliated with government institutions and tiny numbers with corporations. We defined them as non-higher education institution (non-HEI) scientists. To be registered in the SINTA database, a scientist must have published at least one scientific article indexed in Scopus, Google Scholar, or Web of Science. Currently, 233,564 scientists affiliated with 3425 institutions are registered in SINTA (accessed 26 February 2022). Each scientist has a SINTA score calculated using a specific formula [63]. There are at least 86 criteria for defining a scientist's SINTA score, such as the number of written scientific articles or books, the number of copyrights or patents produced, or the number of supervised students. There are scores relative to any research activities.

Before selecting the participants, we filtered the scientists first. Then, we stratified the results by institution since many factors relevant to the study are likely to vary more across institutions than within. Finally, at most ten scientists for each institution were selected randomly (or at least approximately so). Part of this selection was performed not entirely with a random number generator, but only by hand, obtaining participants from each institution with no particular pattern. Since this selection was presumably blind to the characteristics of these scientists, it should not have introduced significant biases in the sample.

We employed two filters to obtain a maximized sample of participants. First, we filtered scientists who had not actively conducted research in the past three years. These scientists included those that were retiring, those that were assigned to other duties, or those that were choosing other professions. They might not be aware of their affiliations' latest regulations. Therefore, we excluded scientists with a 3-year SINTA score of 0. As a result, the population's number was reduced to 171,019 and affiliated with 3155 institutions. We also mapped scientists' departments or expertise subjects into four scientific fields: physical, life, health, and social sciences[9]. Our mapping results showed that more than 50% of the population were social scientists. Thus, we excluded scientists in social and humanities fields, since WMD-applicable research primarily concerns scientists in non-social-science fields. However, the filtering method was imperfect, since the mapping was performed manually. The population's number was again reduced to 85,041 scientists and affiliated with 2014 institutions.

Having identified the population of interest, finally, we selected 8814 participants using a non-proportional random sampling strategy, since we were interested in variabilities at the organization level. We acquired the targeted emails from their published articles. However, emails were sent successfully to 7686 scientists that were affiliated with 1934 institutions. The population and sample target distributions are shown in Table 1.

**Table 1.** Population and sample target.

| Institution Types | Number of Scientists | | Number of Affiliations | |
|---|---|---|---|---|
| | Population | Target | Population | Target |
| "A" or "excellent" HEIs | 77,516 | 798 | 99 | 89 |
| "B" or "very good" HEIs | 100,609 | 2982 | 883 | 707 |
| "C" or "good" HEIs | 37,632 | 2032 | 1503 | 714 |
| Not accredited HEIs | 12,251 | 1639 | 858 | 378 |
| Government and corporate institutions | 5556 | 235 | 82 | 46 |
| Total | 233,564 | 7686 | 3425 | 1934 |

*3.2. Research Instruments*

The questionnaires were divided into three parts. In the first part, we determined respondents' willingness to answer some personal data questions, such as gender, age, and last education level, as shown in Table A1. In the second part of the questionnaire, we asked respondents to answer a few questions about their understanding and experiences of research data governance, as shown in Table A2. A five-point Likert scale was used in the third part of the questionnaire. We requested respondents' willingness to give a weight of 1–5 to the statements, as shown in Table A3, where 1 = strongly disagree, 2 = disagree, 3 = doubtful/do not know, 4 = agree, and 5 = strongly agree.

*3.3. Analysis Methods*

A group of statistical methods in the form of structural equation modeling (SEM) was used to quantify and examine the connections between latent and observable variables. It explores linear causal links among variables while concurrently taking measurement errors into account, making it similar but more effective than regression analyses [64]. In addition, it offers a flexible framework for creating and studying complex interactions between numerous variables, enabling researchers to use empirical models to check the theory's viability.

The SEM with partial least squares (PLS) was performed to analyze the data. PLS-SEM was selected because it can handle many independent variables, even though multicollinearity exists between them. SEM is a multivariate statistical analysis technique used to analyze structural relationships that represent hypotheses in the research model [65]. Meanwhile, PLS is a powerful analytical method because it can be applied to all scale data. Moreover, it does not require many assumptions. The PLS-SEM combines factor analysis and multiple regression analysis, and it is used to analyze the relationships between measured variables and latent constructs. The relationships are used to interpret the hypothesis test's results in determining which factors influence scientists performing research data governance activities.

Several indicators for determining validity and reliability at the model's measurement analysis stage are composite reliability (CR), average variance extracted (AVE), factor loading, and Cronbach's alpha coefficient. Cronbach's alpha is a measure of internal consistency—that is, how closely related a set of items is as a group. It is considered to be a measure of scale reliability. Composite reliability (also called construct reliability) is a measure of internal consistency in scale items, much like Cronbach's alpha. AVE is a coefficient that describes the variance in indicators that common factors can explain. The loading factor is the correlation between the indicator and its latent construct. In many social studies, a construct's measurement is often performed indirectly by using its indicators. Variables and indicators are considered valid and reliable if the parameters are as follows: CR > 0.7, AVE > 0.5, factor loading > 0.5, and Cronbach's alpha coefficient > 0.7 [66].

## 4. Results

Table 2 shows the demographic information of the respondents. The response rate of our survey was 14.88%, among which, 1044 respondents came from HEIs and 74 came from non-HEI. There were also 26 respondents from unaccredited HEIs, but they were not analyzed further because the number of samples was not representative. In general, there were more male than female respondents. Most respondents were less than 40 years old (59%), had obtained a master's degree (74.1%), and had worked for 0 to 10 years as researchers (62.2%). Moreover, most respondents' scientific fields were the physical sciences (43.8%).

**Table 2.** The respondent's demographic characteristics.

| Demographic | | Main Affiliation | | |
|---|---|---|---|---|
| | | **HEIs** | **Non-HEI** | **Total** |
| Gender | Female | 41.4% | 45.9% | 41.7% |
| | Male | 58.6% | 54.1% | 58.3% |
| Age | 20–30 years old | 10.6% | 5.4% | 10.3% |
| | 31–40 years old | 49.6% | 36.5% | 48.7% |
| | 41–50 years old | 25.2% | 29.7% | 25.5% |
| | 51–60 years old | 12.2% | 17.6% | 12.5% |
| | 61–70 years old | 2.4% | 10.8% | 3.0% |
| Last Education | Bachelor's degree | 0.4% | 9.5% | 1.0% |
| | Master's degree | 75.5% | 54.1% | 74.1% |
| | Doctorate's degree | 24.1% | 36.5% | 25.0% |
| Accreditation of the HEIs | A or excellent | 13.4% | 0.0% | 12.5% |
| | B or very good | 53.7% | 0.0% | 50.2% |
| | C or good | 32.9% | 0.0% | 30.7% |
| | Non-HEI | 0.0% | 100.0% | 6.6% |
| Research experience | 0–10 years | 64.1% | 35.1% | 62.2% |
| | 10–20 years | 27.9% | 41.9% | 28.8% |
| | 20–30 years | 6.1% | 9.5% | 6.4% |
| | 30 years | 1.9% | 13.5% | 2.7% |
| Scientific field | Health sciences | 26.0% | 20.3% | 25.6% |
| | Life sciences | 18.6% | 29.7% | 19.3% |
| | Physical sciences | 43.8% | 40.5% | 43.6% |
| | Social sciences | 11.7% | 9.5% | 11.5% |

We found a difference in the length of research experience between the HEIs and non-HEIs. Cumulatively, most scientists in Indonesia are still at the beginner level (more than 60% had experience under ten years). The research experience of non-HEI respondents appeared to be longer than that of HEIs. This is because the number of scientists aged above 61 years in non-HEIs was higher than that in HEIs. From the last educational background of the respondents, it was clear that there were still scientists with undergraduate backgrounds. Non-HEIs had more scientists with an undergraduate background than HEIs.

Most respondents (78.7%) had access to digital materials. As depicted in Figure 2, Indonesian scientists tend to collect data in the course of their own research activities in collaboration with colleagues at their department. Furthermore, they stored and processed the data using personal infrastructure, but actively shared it with colleagues in the department or research center primarily by using emails. Only a portion of them used a cloud-based external server to store their data (<30%) or processed their data using a cloud-based external application (about 20%) and minimally shared the data with the public.

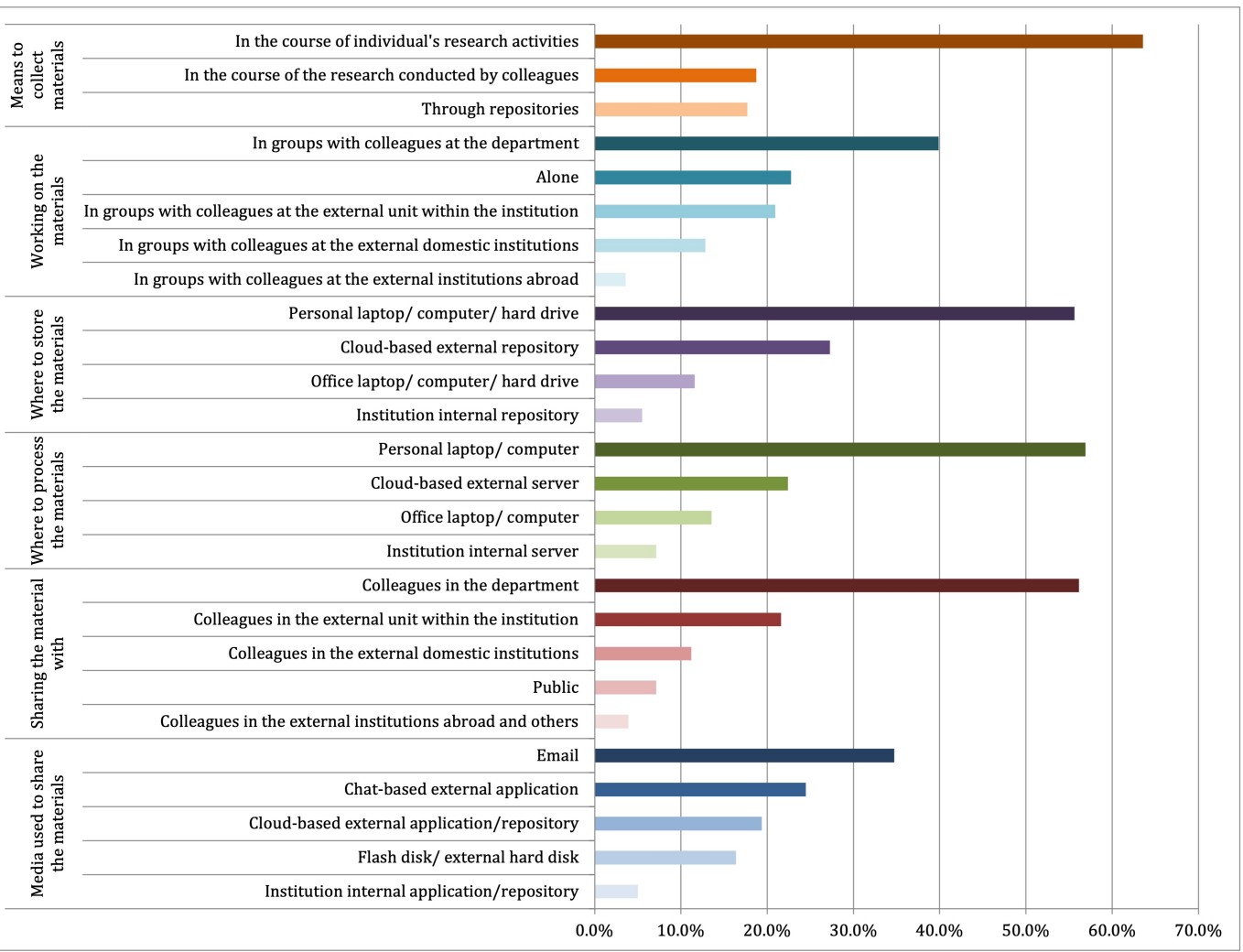

**Figure 2.** Indonesian scientists' behavior in producing, storing, accessing, or sharing digital materials.

Figure 3 shows the respondents' practices in sharing digital materials. While 15.9% of the respondents thought that their materials could not be shared with other parties, 26.9% never shared their materials. Subsequently, 2.5% of respondents thought that their materials could be shared with other parties, and only 0.5% always shared their materials. Furthermore, they sometimes warned receivers to keep the materials safe and secure, and this had a scale mean of 3.03.

Moreover, Figure 4 shows the respondents' practices in handling digital materials. When retrieving information from a repository, 41.6% of respondents always read the terms and conditions, and this had a scale mean of 3.82. On the other hand, 33.4% of respondents also always read the terms and conditions when storing materials into a repository, and this had a scale mean of 3.68. Nevertheless, 31.1% of respondents never locked documents or folders, and this had a scale mean of 2.54.

Based on Indonesian scientists' practices in producing, storing, accessing, and sharing and handling digital materials, we examined their perceptions about the safety of materials, as depicted in Figure 5. As a result, 33.4% of respondents often felt that the means (media) used to share the materials were adequately safe, and this had a scale mean of 3.38. On the other hand, 30.5% of respondents often thought that the means (media) used to store the materials were safe, and this had a scale mean of 3.45. Nevertheless, 33.4% of respondents also felt that unauthorized parties could probably have access to and improperly use some or all of their materials, and this had a scale mean of 3.25.

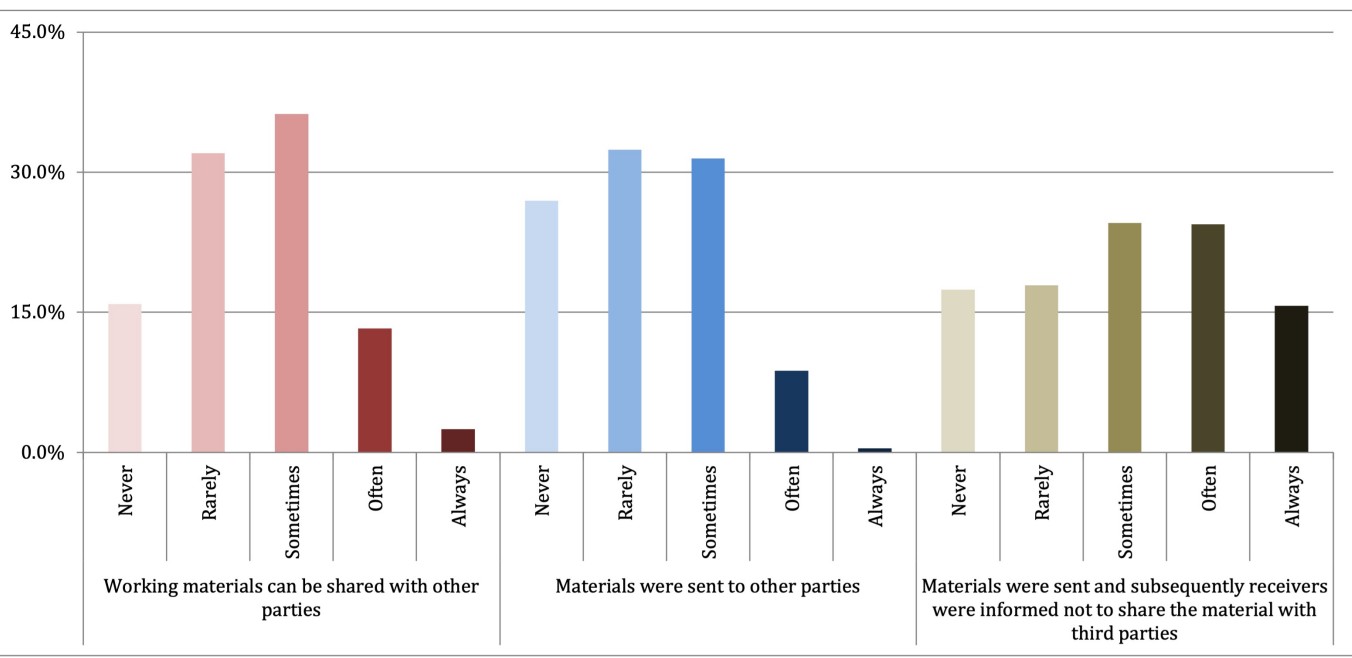

**Figure 3.** Indonesian scientists' behavior in sharing digital materials.

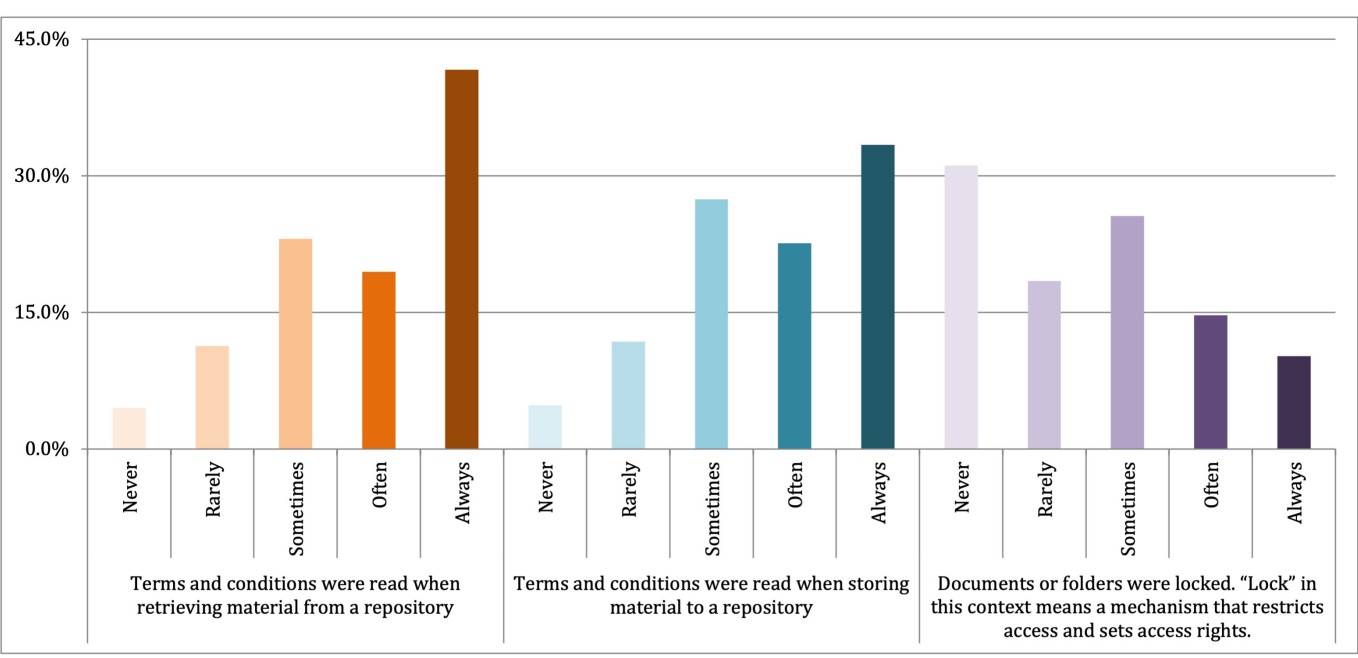

**Figure 4.** Indonesian scientists' behavior in handling digital materials.

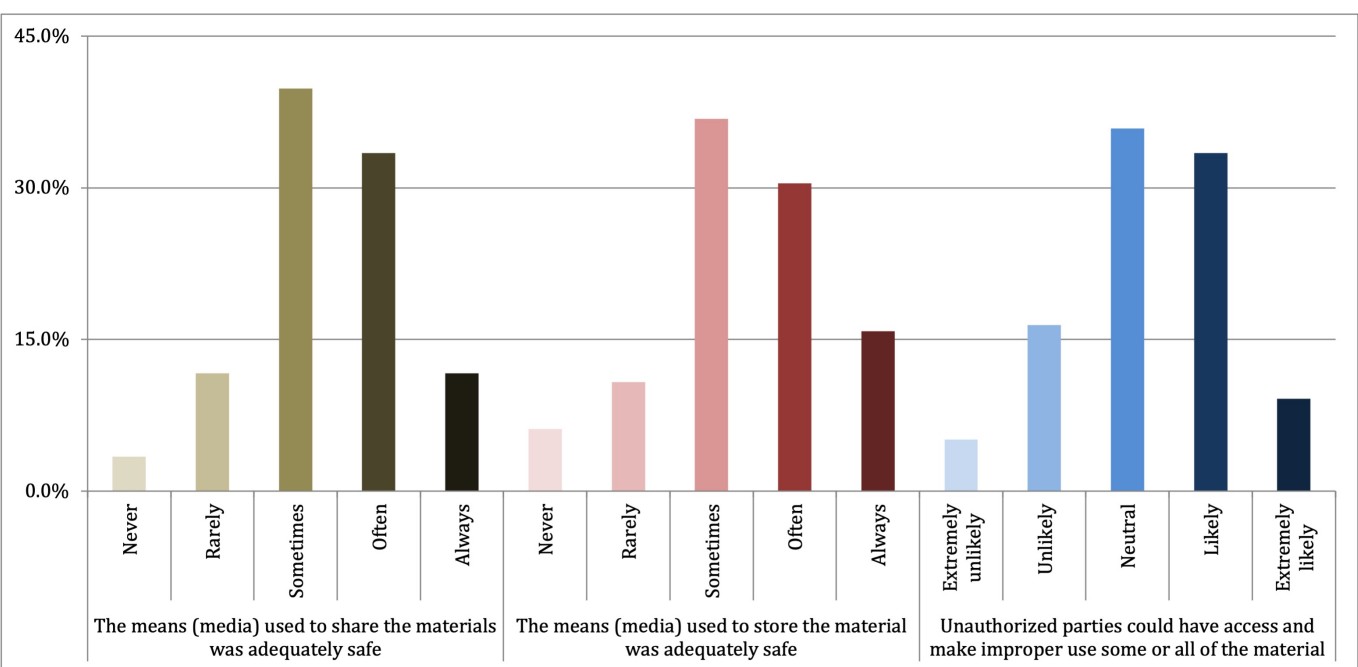

**Figure 5.** Indonesian scientists' perceptions of the safety of digital materials.

Finally, we examined the perceptions of respondents regarding the existence of their institutions' policies and standards related to research data governance, as shown in Figure 6, and their level of knowledge of the institution's procedures and the best practices in research data governance, as depicted in Figure 7 and Figure 8, respectively. As a result, 35.1% of respondents were at least moderately familiar with best practices in research data governance. Moreover, 45.88% of respondents knew that their institutions have policies related to research data governance. Furthermore, among those respondents, 56.2% of them were at least moderately familiar with the policies.

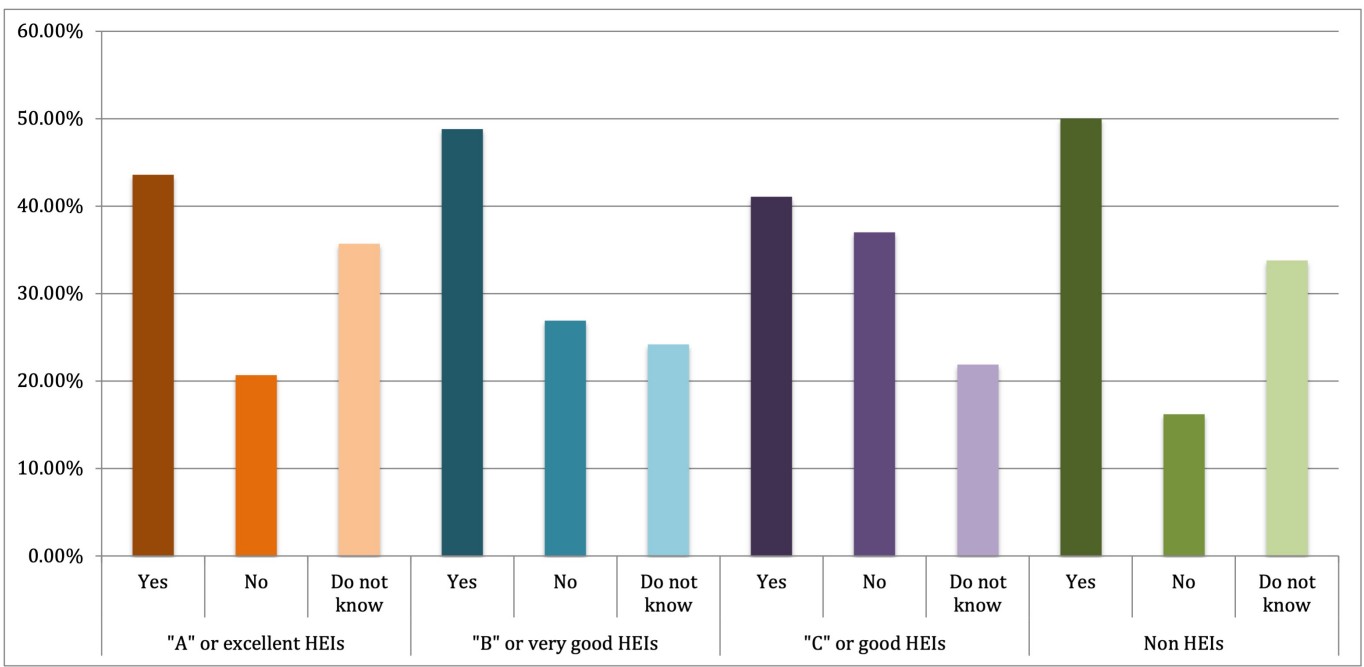

**Figure 6.** Indonesian scientists' perceptions of the existence of their institutions' policies and procedures related to research data governance.

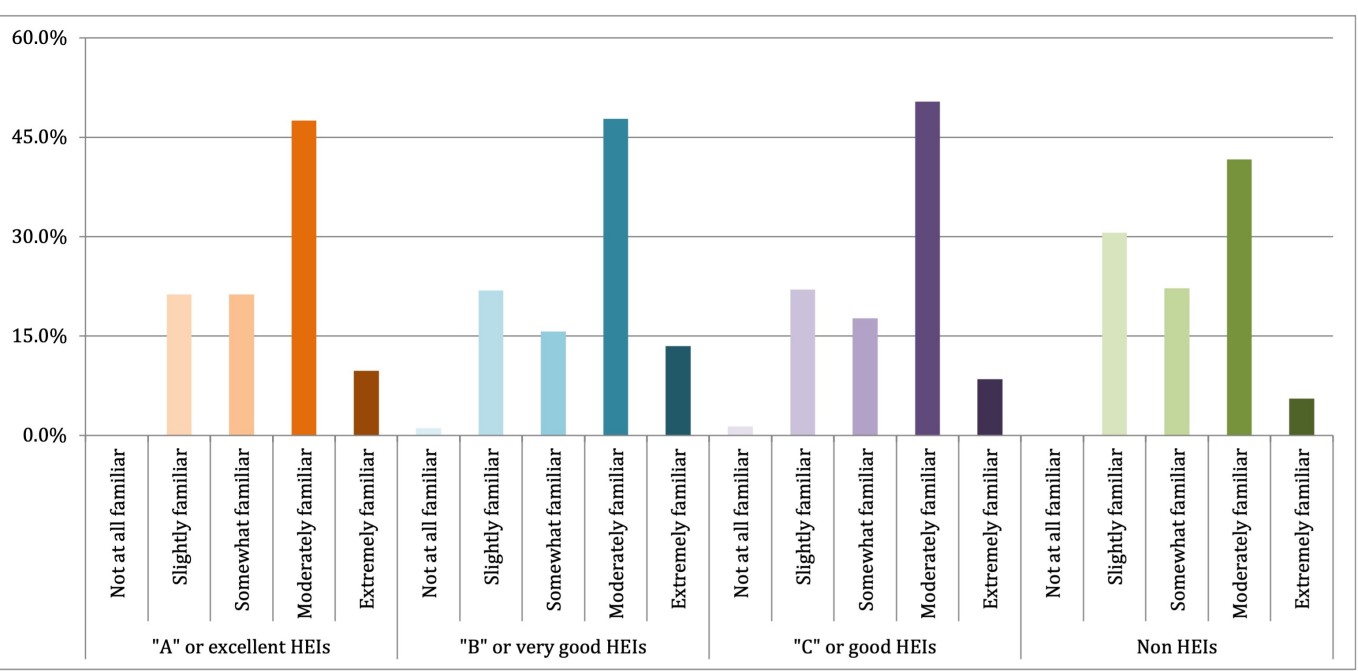

**Figure 7.** Indonesian scientists' perceptions of their level of knowledge of institutions' policies or procedures regarding research data governance.

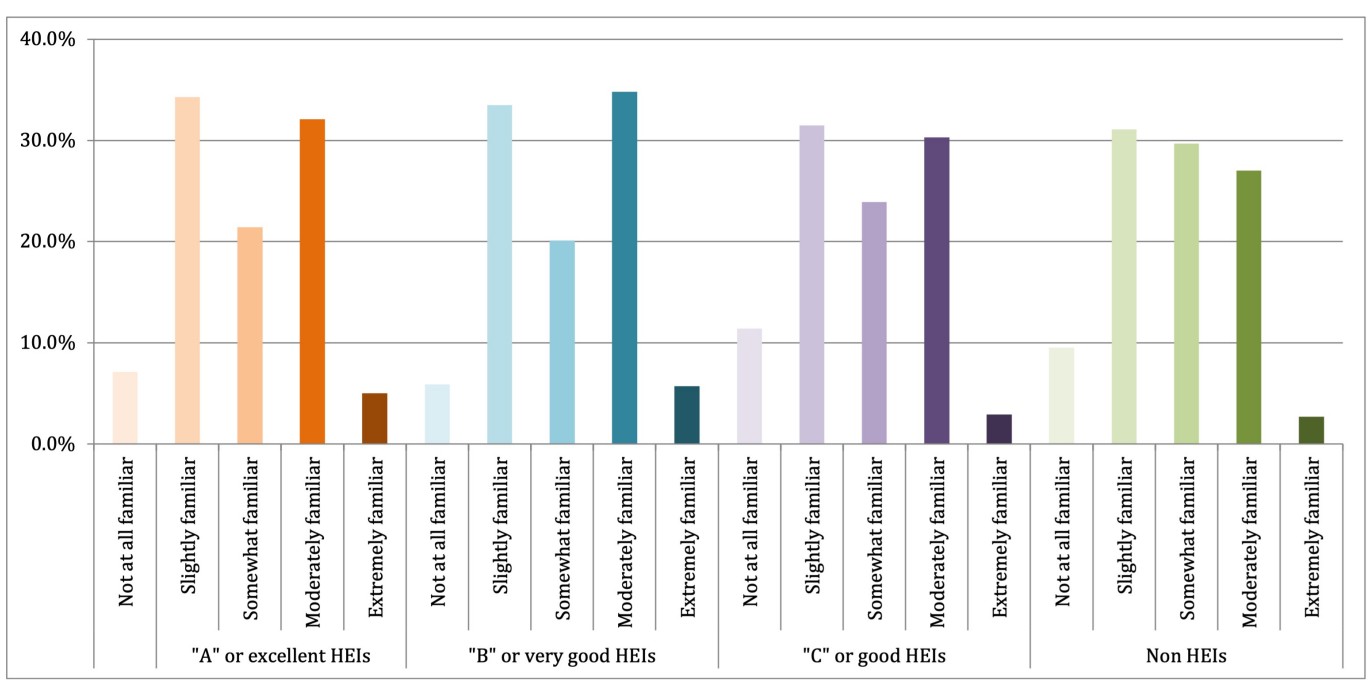

**Figure 8.** Indonesian scientists' perceptions of their level of knowledge on the best practices in research data governance.

Before performing the SEM analysis, the model was measured first using validity and reliability tests. Table 3 shows the analysis's results of the final measurement model. It shows that all variables included in the structural model analysis fulfilled the validity and reliability requirements. The values of AVE and factor loading were higher than the minimum of 0.5 to meet validity requirements. Meanwhile, the values of CR and Cronbach's alpha coefficients were higher than the predetermined level of 0.7 to satisfy the reliability requirement.

**Table 3.** The descriptive statistics and validity and reliability test results of the measurement model.

| Variable | Indicator | Mean | Std Dev | CR | AVE | Factor Loading | Cronbach's Alpha Coefficient |
|---|---|---|---|---|---|---|---|
| WMD and Data Governance Awareness | aw1 aw2 aw3 aw4 aw5 | 3.68 | 0.63 | 0.83 | 0.50 | 0.59 0.67 0.78 0.80 0.69 | 0.75 |
| Data Governance Organization Policy | op1 op2 op3 op4 op5 | 3.81 | 0.58 | 0.84 | 0.52 | 0.74 0.79 0.68 0.81 0.54 | 0.76 |
| Data Governance Experience and Involvement | ei1 ei2 ei3 ei4 ei5 ei6 | 3.74 | 0.61 | 0.90 | 0.60 | 0.78 0.70 0.83 0.77 0.82 0.75 | 0.87 |
| Attitude | att1 att2 att3 att4 att5 | 4.20 | 0.55 | 0.91 | 0.67 | 0.84 0.88 0.88 0.71 0.77 | 0.87 |
| Subjective Norm | sn1 sn2 sn3 sn4 sn5 sn6 | 3.67 | 0.59 | 0.87 | 0.53 | 0.70 0.84 0.87 0.78 0.56 0.59 | 0.82 |
| Perceived Behavioral Control | pbc1 pbc2 pbc3 pbc4 | 3.98 | 0.53 | 0.86 | 0.61 | 0.73 0.81 0.78 0.79 | 0.79 |
| Threat Appraisal | ta1 ta2 ta3 ta4 ta5 ta6 | 4.26 | 0.53 | 0.90 | 0.59 | 0.74 0.74 0.83 0.79 0.82 0.69 | 0.86 |
| Coping Appraisal | ca1 ca2 ca3 ca4 ca5 ca6 | 3.56 | 0.65 | 0.93 | 0.69 | 0.79 0.82 0.86 0.86 0.80 0.85 | 0.91 |
| Data Governance Behavior | dgb1 dgb2 dgb3 dgb4 dgb5 dgb6 | 3.92 | 0.52 | 0.89 | 0.57 | 0.74 0.77 0.81 0.80 0.71 0.70 | 0.85 |

Table 4 and Figure 9 show the results of the structural model's analysis. From Table 4, it can be observed that the structural model fulfilled the criteria of the goodness of fit. Therefore, the structural model has a good fit. In other words, the proposed independent variables simultaneously affected the data governance behavior of scientists. Based on the *p*-value and the structural equation coefficients of the independent variables, all tested hypotheses of RQ 4 were supported by the obtained data.

**Table 4.** The goodness of fit of the structural model.

| Criteria | Cut-Off Value | Results | References |
|---|---|---|---|
| $\chi^2/df$ | <5 | 5 | [67–69] |
| rms_theta | <0.12 | 0.10 | [70] |
| SRMR | $\leq$0.09 | 0.08 | [71] |

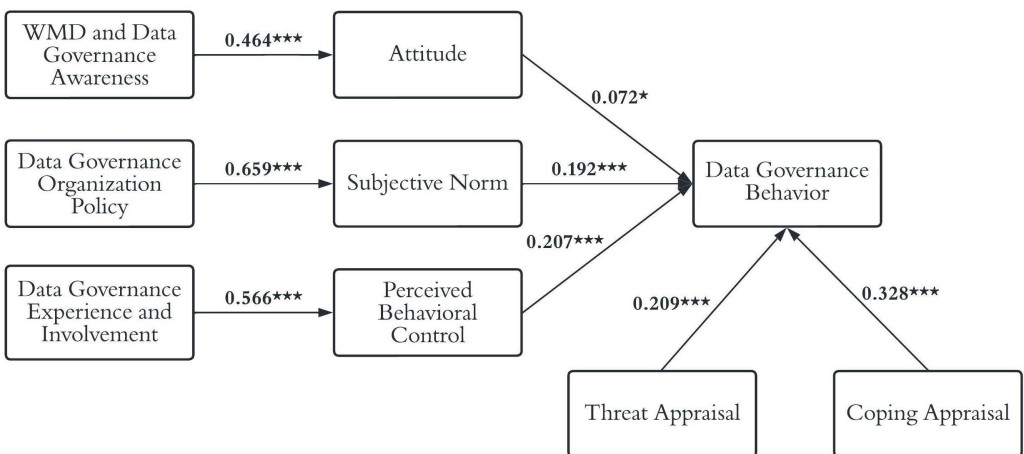

**Figure 9.** The results of the structural model analysis (*, *p* < 0.05; ***, *p* < 0.001).

The findings show that the paths from WMD and research data governance awareness toward attitude ($\beta = 0.464$); from data governance organization policy to subjective norms ($\beta = 0.659$); from experience and involvement to perceived behavioral control ($\beta = 0.566$); and attitude ($\beta = 0.072$), subjective norms ($\beta = 0.192$), perceived behavioral control ($\beta = 0.207$), threat appraisal ($\beta = 0.209$), and coping appraisal ($\beta = 0.328$) toward data governance behavior were significant. Looking further at the effect size for each independent variable, as shown in Figure 9, it can be observed that the independent variable attitude produces the smallest effect size compared to other independent variables. At the same time, the explanatory variable for coping appraisal produces a most significant effect size compared to other variables.

## 5. Discussions

The awareness of research-material-sharing activities seems to have increased, as seen in Figure 3. The result implies that Indonesian scientists might know when to share their materials and when not to. However, some respondents felt that the means (media) they used to share and store digital materials were safe, and this amounted to as much as 11.7% and 15.8%, respectively, as observed in Figure 5. This idea is dangerous, since there are no safe means. In addition to using emails, chat-based external applications, cloud-based external applications or repositories, and flash disks or external hard disks were frequently used to share their research materials. Even institution internal applications or repositories, which are considered safer media, were rarely used. Furthermore, in addition to personal infrastructure such as laptops, computers, or hard drives, cloud-based external repositories and office laptops, computers, or hard drives were frequently used to store research materials. Institutional internal repositories, which are considered safer media, were rarely used. Therefore, to answer RQ 1, even though the awareness of research data governance has started to increase, Indonesian scientists' practices in producing, storing, accessing, or sharing digital materials shape the necessity for practicing research data governance.

Most Indonesian scientists practiced the basic ideas of data governance, such as reading terms and conditions when retrieving materials from a repository or storing materials in a repository, as observed in Figure 4. However, most did not practice advanced practices for securing their materials, such as locking documents or folders to could restrict access and setting access rights. Therefore, to answer RQ 2, even though some basic data governance practices have been practiced, Indonesian scientists' handling skills with respect to research materials should be elevated to meet the standard of good data governance practices. Furthermore, as observed in Figure 5, most believed that unauthorized parties could have access to and improperly use some or all research materials. Thus, this study's results have further highlighted the need for special treatment strategies for data. Moreover,

it is possible to identify vulnerable sources and actions that require data governance strategies.

Furthermore, the patterns in Figures 6 and 7 can be used to answer RQ 3. The pattern indicates that different practices do not exist on the institutional level across different institution types or for accreditation statuses regarding the existence or knowledge of policies, standards, and procedures related to research data governance. Moreover, the same pattern is again shown in Figure 8, which again indicates no difference in practices at the institution level across different institution types or accreditation statuses regarding best practices.

SEM analyses that were performed to answer RQ 4 found that awareness significantly affects attitudes toward data governance behavior. Secondly, data governance organization policies influence subjective norms significantly. Thirdly, the perceived behavioral control was shown to be significantly affected by experience and involvement. Moreover, all native variables of TPB, such as attitude, subjective norm, and perceived behavioral control; and native variables of PMT, such as threat and coping appraisal, significantly influence behaviors.

One of the highlighted results in this study is that coping appraisal was one of the explanatory variables that had the most significant influence on data governance behaviors compared to other explanatory variables. This indicates that the scientists consider complying with data governance recommendations in order to eliminate threats that can be detrimental in the future [28]. Thus, the scientists try their best to practice behaviors contributing to data governance. The more positive the scientist's assessment of coping, the higher the probability that they perform behaviors related to data governance. In addition, it is also known that threat appraisal was the second largest explanatory variable that affects data governance behavior. This phenomenon indicates that scientists consider it necessary to assess the severity and seriousness if they do not perform good data governance behavior [28,72]. These two explanatory variables are native variables of PMT. Therefore, it indicates that PMT is a theory that best explains data governance behavior.

Nevertheless, another highlighted result in this study is the small effect of the attitude on data governance behavior. Scientists might have good attitudes, as they tend to be high-scoring, and the mean of indicators equals 4.2 (as seen in Table 3), but the attitudes have low influences on data governance behavior. One possible reason for this finding might be because the respondents have a low perception of the knowledge needed for data governance both with respect to themselves and their institutions, as observed in Table 5.

**Table 5.** Indonesian scientists' level of knowledge.

| Level of Knowledge (1–5) | Non-HEIs | | HEIs | | Total | |
|---|---|---|---|---|---|---|
| | Mean | Std Dev | Mean | Std Dev | Mean | Std Dev |
| Research data governance best practices | 2.82 | 1.03 | 2.94 | 1.08 | 2.93 | 1.07 |
| Institution's policies, standard, or procedures regarding research data governance | 3.22 | 0.96 | 3.48 | 0.99 | 3.46 | 0.99 |

To assess this hypothesis, we performed two secondary analyses. The first was a SEM analysis performed to examine the effect of the best practices of research data governance only, and the other one was performed to investigate merely 45.88% of respondents who knew that their institutions have policies, standards, or procedures regarding research data governance, which amounted to 513 results with respect to the obtained data. In the first analysis, we divided all data results into two groups, namely, K1a and K1b. The first group, K1a, comprised data from respondents with low knowledge of research data governance practices, and the second group, K1b, included data from highly knowledgeable respondents. Since the mean of the knowledge of best practices was 2.93, K1a consisted of

respondents who chose a value of one or two, which amounted to 456 results with respect to the data, and K1b contained respondents who chose a value of three, four, or five, which amounted 662 results with respect to the data.

In the other analysis, we divided the data into two other groups: K2a consisted of respondents with insufficient knowledge of the institution's policies, and K2b contained the data with respondents with high knowledge. Since the mean level of the knowledge of institution policies is 3.46, respondents who chose a value of one, two, or three were included in K2a, which amounted to 209 results with respect to the data. In contrast, respondents who chose a value of four or five were included in K2b, which amounted to 304 results with respect to the data. The results of the analyses are shown in Figure 10.

The attitude has much weaker effect on data governance behavior in groups with a low level of knowledge, such as K1a and K2a, since the $p$-value is higher than 0.05. On the contrary, the attitude significantly affects data governance behavior in groups with a high level of knowledge, such as K1b and K2b with coefficients of 0.1 and 0.137, respectively. Thus, these results support our hypothesis that the cause of the small influence of attitudes toward data governance behavior in total is the low level of knowledge of Indonesian scientists with respect to best practices in research data governance and their institutions' policies or procedures regarding research data governance. In other words, the low influence of attitudes on behavior has been proven to be caused by scientists' lack of knowledge with respect to data governance (both themselves and institutionally).

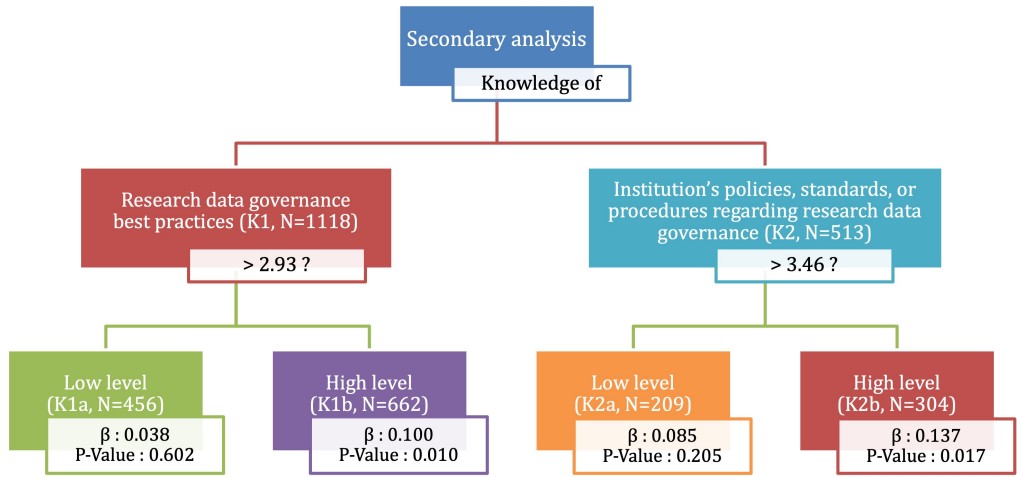

**Figure 10.** Secondary analyses of the effect of attitudes on data governance behavior for groups with different levels of knowledge.

### 5.1. Research Implications

Our findings in answering RQ 1, RQ 2, and RQ 3 strengthen the importance of this study. If these findings are common across other countries similar to Indonesia, the implication is that similar studies and interventions are also necessary in these countries. For example, from a quantitative survey of life scientists in 13 countries in Sub-Saharan Africa [73], the scientists shared their data via emails, institutional repositories, online databases, or even personal web pages. Moreover, the personal connections between scientists affected their confidence in sharing data. More than half of the respondents were comfortable sharing pre-publication data with other familiar individuals. The main motivations for sharing data include improving research visibility and expanding collaborations. A survey of 1372 respondents across 116 countries in the geophysicists' community revealed that they also have concerns about potential data misuse and improper citation or acknowledgment [74]. They also tended to generate or collect the data that a research team member or colleagues collect. They were also willing to share data across a broad group of researchers, or they were willing to place it in a central repository. Furthermore, most respondents reported that their organizations do not provide training or assistance

in data management practices in some disciplines. In line with our findings, Indonesian scientists tend to collect data alone or with research team colleagues. In addition, they tend to share data with colleagues primarily using email or online databases. Even though the positive contributions of organization support in providing policies and procedures relative to research data governance have been reported, most (more than 50%) respondents (regardless of their institution accreditation) reported opposing contributions or reports being uninformed.

The first finding of the answers for RQ 4 is in line with studies that used awareness determinants in various contexts [59,75,76]. This indicates the importance of increasing awareness, since it has a significant indirect effect on the intended behavior. Therefore, the awareness variable could be used in other domains as well. The second finding corroborates studies in the information security (IS) domain [59,77], where IS security organization policies were used as determinants. This implies that policies that should be followed in organizations that can positively create mandatory conditions to perform the intended behavior. As confirmed by [59], the third finding implies that more experience in practicing the intended behavior will likely lead to a higher frequency of exhibiting that behavior. Furthermore, TPB and PMT were sufficient for explaining the factors determining Indonesian scientists' likelihood of practicing data governance to prevent WMD-applicable technology transfer. These findings are supported by studies in [49–55,58–60].

### 5.2. Practice Implications

The findings in this study suggest that organizations or practitioners should regularly increase scientists' awareness regarding the WMD-applicable technology transfer and research data governance, since the awareness positively influences attitudes toward data governance behavior. Our results indicated that scientists have started to become aware of data governance practices. However, most perceived that their means (media) of storing and sharing their materials tend to be safe. One possible reason for this idea could be a deficiency in the knowledge of research data governance best practices. Of course, the means are never safe, but precautions or special treatments on the materials make them secure. Therefore, lectures or webinars about the best practices of research data governance should be held frequently to increase scientists' knowledge about data governance behavior and raise their awareness. Based on our findings, knowledge is also an influencing factor in scientists' attitudes toward data governance behavior. The more awareness and knowledge respondents possess, the more positive their attitudes toward the intended behavior.

Moreover, this study found that data governance organization policies affect subjective norms significantly toward data governance behavior. Nevertheless, less than 50% of scientists were sure that their organizations have data governance policies. Before conducting the survey, we hypothesized that there might be differences in data governance practices at the institution level across institution types and accreditation levels. However, the results showed that no differences were found. Therefore, research institutions or higher authorities are encouraged to establish data governance regulations and to promote them regularly. The more people behave consistently according to the policies, the more probable it is that other people will be influenced toward practicing data governance. Governing data does not mean necessarily closing the data but giving access to those eligible to open the data. If scientists need to open the data publicly, they must ensure that the research data are safe and secure, not WMD-applicable. Research data governance rules these measures. For example, research institutions are encouraged to establish a structured organization of data governance, where some roles perform multi-level approvals to provide a thorough review process. The organization should be tailored according to the institution's needs, since these are likely to vary by country and possibly field of research.

Research data management (RDM) consists of several activities and processes associated with the life cycle of data, including acquisition, storage, security, preservation, retrieval, sharing, and reuse based on ethical considerations, legal issues, and governance frameworks. RDM is common in developed countries but is a relatively new concept in

developing countries, including Indonesia, where national data governance policies are still in their early stages. Indonesian law number 11 of 2019 is the highest legal instrument that regulates RDM in Indonesia. However, it only governs the storage regulations when funders, scientists, and institutions must store the research data in an integrated system. Meanwhile, other activities of RDM still need to be governed. Government regulations derived from the law have yet to be made available to this day, and policies governing RDM are established by the respective institutions.

This study also found that experience and involvement significantly affected the perceived behavioral control toward data governance behaviors. Nonetheless, our results showed that even though some scientists have been practicing some basic ideas of data governance, scientists' material handling skills should be elevated. For instance, hands-on training or workshops should be conducted. Scientists participating in these processes can learn how to protect their materials by restricting access and setting access rights by using simple lock mechanisms and advanced encryption techniques. Furthermore, the more experience and involvement scientists have, the easier it is for them to practice daily data governance behavior. From the literature, it is clear that special treatments for data are required in different use cases to be used properly while simultaneously preventing data misuse. For example, several preventive measures are necessary when sharing or storing data, including data protection from unwanted access; cryptographic techniques, such as blockchain, for safeguarding; and data obfuscation for privacy preservation.

*5.3. Limitations and Future Works*

There are some limitations of this study. First, the large sample size of this study is a strong advantage, but there is likely to be a bias in the responses (or at least unknown factors). Furthermore, the response rate was sufficiently high but certainly not high, which might raise concerns about the study's internal validity. Moreover, the dropout rate was 42.31%—839 respondents did not finish the survey. One possible reason could be that tokens of appreciation, such as monetary rewards or others, were not given to the participants due to the limited budget or regulations, thereby making them demotivated to answer all questions. Moreover, some candidates, out of 7686 participants, did not receive emails because their addresses were invalid or errors occurred when sending the emails. In total, 400 emails were soft-bounced and 728 were hard-bounced.

By using an a priori approach [78], the required sample sizes for the SEM were 133 and 264 samples for the model's structure and detecting effects, respectively, given the number of latent variables, 9; the number of indicator items, 49; the anticipated effect size of 0.3 (medium); the statistical power level of 0.95; and a probability level of 0.05. Meanwhile, considering the population, the minimum sample size was 348, according to Slovin's formula [79]. Therefore, the response rate of 14.88%, which amounted to 1118 respondents, was considered a sufficient number. However, it would be preferable if the response rate were to be higher, since it would mean more research institutions would be catered to in the study.

Many other indicators might influence the pattern of data governance behavior for academics because the factor analysis carried out in this research was based on confirmatory factor analyses. For example, there may be differences in the behavioral patterns of data governance in academia due to the experience of conducting international collaborative research. Different nations have different policies for managing data, what type of data may be shared, and with whom the data are shared. Participants' interests, such as data openness, could also influence the analysis. Future work might explore how data governance practices may vary depending on individual scientists' collaboration patterns, research field, and other factors that this study was not designed to assess.

This study used PLS-SEM, but the construction of the model was only based on a collection of previous related theories. PLS-SEM allows the exploration of relationships between variables that were not present in previous findings so that further research can explore this area (when the theoretical basis of the construct or model is still weak) [80].

For example, awareness is related to attitude and its direct influence on data governance behavior in addition to exploring other analytical factors. One weakness in using PLS-SEM is that it only functions as a predictor analysis tool and not as a test for the model [80]. Thus, in order to explore the model further, it is better to use covariance-based SEM (CB-SEM).

Differences in several demographic variables in this study, such as research experience, educational background, and age distribution, could indicate a disparity in the structure's model between HEIs and non-HEI scientists. For example, the age difference could inevitably impact the cognitive abilities of scientists in learning new things [81]. Older scientists show significantly less learning progression than younger ones. However, the SEM analysis for non-HEI could not be performed, since the model did not fit when only using data from non-HEI respondents in this study. Although the response rate of non-HEI respondents was higher than others, the number was probably a bit low for a sample for SEM. Nevertheless, it might also indicate the differences between HEIs and non-HEI structural models. Future research could further elaborate on these differences.

Moreover, the response rate of unaccredited HEIs was under-represented, as shown in Figure 11. One possible reason for this limitation was that the respondents' actual data might not be the same as those in our database. For example, a target's affiliation was not accredited in our database, but in the survey, the respondent answered "A" or "B" or "C" on the affiliation accreditation question. Therefore, it could be that either the accreditation data or the respondent's affiliation was not updated in our database. There was also a small chance that respondents did not know the accreditation levels of their affiliations and inserted random values. However, this is improbable, since we provided a link for them to check their institutions' accreditation levels when answering the question.

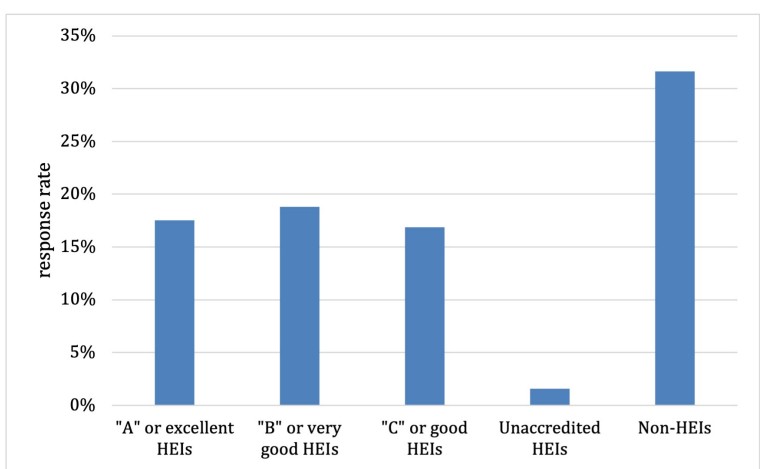

**Figure 11.** Response rate of the survey.

Besides increasing the number of respondents of non-HEI and unaccredited HEIs, future work also could consider the mediating intention variable. This is because the scientists might want to manage data well but are constrained by their facilities and organizational policies when adequately carrying out research data governance. It is known that the intention variable can explain 28% of the variance, on average, with respect to future behavior [82].

**Author Contributions:** Conceptualization, L.P.M. and Z.A.; methodology, L.P.M., Z.A. and A.Y.; software, L.P.M. and A.Y.; validation, L.P.M., Z.A. and A.I.; formal analysis, A.Y.; investigation, A.I.; resources, L.P.M.; data curation, L.P.M. and A.Y.; writing—original draft preparation, L.P.M.; writing—review and editing, Z.A., A.Y. and A.I.; visualization, L.P.M.; supervision, Z.A.; project administration, L.P.M.; funding acquisition, L.P.M., Z.A. and A.I. All authors have read and agreed to the published version of the manuscript.

**Funding:** This research was funded by CRDF Global grant number 202109-68097.

**Institutional Review Board Statement:** The study was conducted according to the guidelines of the Declaration of Helsinki and approved by the Institutional Review Board (Research Ethical Committee on Social Sciences and Humanities) of the National Research and Innovation Agency of Republic of Indonesia (no. 87.a/Klirens/XI/2021, 17 November 2021).

**Informed Consent Statement:** Informed consent was obtained from all subjects involved in the study.

**Data Availability Statement:** The data generated during this study and those supporting the re-ported results can be found at https://hdl.handle.net/20.500.12690/RIN/E6SHXB (accessed on 27 September 2022).

**Acknowledgments:** The authors gratefully thank Daniele Fanelli for helpful advice on various aspects of this study, including the study's design, data analyses, the interpretation of results, and publication.

**Conflicts of Interest:** The authors declare no conflict of interest. The funders had no role in the design of the study; in the collection, analyses, or interpretation of data; in the writing of the manuscript; or in the decision to publish the results.

## Appendix A

*Appendix A.1*

In the first part of this questionnaire, we examined respondents' willingness to answer some personal data questions.

**Table A1.** Questionnaire: part one.

| No | Question |
|---|---|
| 1 | Age |
| 2 | Gender |
| 3 | Last education |
| 4 | Research experience |
| 5 | Main affiliation |
| 6 | If main affiliation is a university, accreditation of the affiliated university; if you are unsure about choosing one of the options, please refer to https://pddikti.kemdikbud.go.id (accessed on 1 March 2022) |
| 7 | Scientific field; if you have doubts about choosing one of the options, please refer to: https://service.elsevier.com/app/answers/detail/a_id/15181/supporthub/scopus (accessed on 1 March 2022) |

*Appendix A.2*

In the second part of this questionnaire, we examined respondents' willingness to answer a few questions about their understanding and experiences of research data gover-nance. The data in this context comprise data obtained during research activities and not those used in the pre-study (e.g., literature for the literature review, ICP, and proposals), such as the following:

- Scientific experiment;
- Models and simulations, e.g., models with associated metadata and computational data arising from the models;
- Observation, namely, a particular phenomenon at a specific time or location;
- Interviews;
- Survey.

Any such types of data or similar data will be collectively referred to in the survey as "materials," which includes but are not limited to the following forms:

- Notes, graphs, tables, maps, images, videos, audio, or visual recordings;
- Voice recordings and transcripts of interview activities;
- Derived data, namely, data that results from processing and combining raw data;
- Canonical or reference data, e.g., gene sequences, chemical structures, and others;

- Materials that accompany research activities, including coding instructions, interview instructions, flow charts of data collection, questionnaires, information on research methods and techniques used, code books, data collection instruments, statistical summaries, database dictionaries, summary/description of activities, and bibliographies of publications related to the data;
- Scientific articles or technical/research reports that are the outputs of research activities.

**Table A2.** Questionnaire: part two.

| No | Question | Control form |
|---|---|---|
| 1 | Do you have access to any of the materials listed above? "Access" entails the potential to view data. It does not necessarily entail working with the data. | If "No" was chosen, jump to question no. 17. |
| 2 | By what means is this material collected? | If there was no answer, jump to question no. 6. |
| 3 | Do you work on the material, and with whom? | |
| 4 | Where do you process the material? | |
| 5 | Where do you store the materials (either the raw materials or the results of the material processing)? | |
| 6 | In your opinion, how often can the material you are working with be shared by other parties? | If "Never" was chosen, jump to question no. 11. |
| 7 | Have you ever shared or sent the material to other parties? | |
| 8 | With whom did you share the material? | |
| 9 | What media did you use to share the material? | |
| 10 | In your opinion, how often do you feel that the means (media) you have used to share the materials was adequately safe, given the nature of the material? "Safe" in this context means that only the intended receivers can have intended access to the materials. | |
| 11 | How often have you sent material and subsequently informed the receivers not to share the material with third parties? | |
| 12 | When retrieving material from a repository, do you read the terms and conditions? | |
| 13 | When storing material to a repository, do you read the terms and conditions? | |
| 14 | In your opinion, how often do you feel that the means by which you store the material is adequately safe, relative to the nature of the material? By "safe", we mean that no unintended parties have unintended access to the materials. | |
| 15 | In your opinion, how likely is it that unauthorized parties could have access and make improper use some or all of the material you currently have? | |
| 16 | Have you ever locked a document or folder? "Lock" in this context means a mechanism that restricts access and sets access rights. | |
| 17 | What is your level of knowledge of research data governance best practices? | If "No" or "Do not know" was chosen, jump to Part 3. |
| 18 | To your knowledge, does your institution have policies and procedures related to research data governance? | |
| 19 | What is your level of knowledge of your institution's policies or procedures regarding research data governance (data deposition, disclosure, access, use, and data preservation)? | |

**Table A3.** Questionnaire: part three.

| Variable | Code | Description of Indicator | Adopted From |
|---|---|---|---|
| WMD and Data Governance Awareness | aw1 | I am aware that there is a potential threat of misuse of research data being used as a WMD. | [59] |
| | aw2 | I have sufficient knowledge about the consequences of misuse of research data as a WMD. | |
| | aw3 | I understand the risk of incidents of access to research data by WMD those who could use it as a WMD. | |
| | aw4 | I continue to update myself regarding data governance awareness in preventing misuse of research data from being applied to become WMD by irresponsible parties. | |
| | aw5 | I share knowledge of data governance with my colleagues in preventing misuse of research data from being used as a WMD to raise my awareness. | |
| Data Governance Organization Policy | op1 | Research data governance policies and procedures are central to my organization. | [59] |
| | op2 | Research data governance policies and procedures influence my behavior. | |
| | op3 | Research data governance policies and procedures caught my attention. | |
| | op4 | Behavior that conforms to research data policies and governance is a value in my organization. | |
| | op5 | I have sufficient knowledge about government policies related to data governance. | |
| Data Governance Experience and Involvement | ei1 | Experience in research data governance enhances my ability to behave safely. | [59] |
| | ei2 | I am actively involved in the governance of research data, and I care about my behavior in my work. | |
| | ei3 | Experience helps me identify and assess threats from unwanted parties accessing protected research data. | |
| | ei4 | I can sense the threat of research data misuse because of my experience. | |
| | ei5 | Experience helps me to do research data governance. | |
| | ei6 | I have the appropriate ability to manage risks related to research data security due to my experience. | |
| Attitude | att1 | Data governance is needed to prevent and reduce the risk of misuse of research data as a WMD. | [53,55,59] |
| | att2 | Data governance helps provide a sense of security for research results to prevent misuse of research data as a WMD. | |
| | att3 | Practicing data governance is a good idea to prevent misuse of research data from being used as a WMD. | |
| | att4 | I have a positive view of changing the behavior of my colleagues to conduct research data governance. | |
| | att5 | I believe that research data governance is very valuable in an organization. | |
| Subjective Norm | sn1 | Research data governance policies and procedures in my organization are important to my colleagues. | [53,55,59] |
| | sn2 | Research data governance policies and procedures in my organization influence my behavior. | |
| | sn3 | The research data governance culture in my organization influences my behavior. | |
| | sn4 | My supervisor's research data governance influences my behavior. | |
| | sn5 | To date, I have carried out research data governance following the practice carried out by my seniors. | |
| | sn6 | The management of research data that I do cannot be separated from the direction of my superiors. | |
| Perceived Behavioral Control | pbc1 | I believe that research data governance is not a difficult thing. | [55,59] |
| | pbc2 | I believe that experience helps me be careful in conducting research data governance. | |
| | pbc3 | Following research data governance policies and procedures is easy for me. | |
| | pbc4 | Research data governance is a doable practice. | |
| Threat Appraisal | ta1 | It is a serious problem if other parties can access my research data without my consent or knowledge. | [51,53,54] |
| | ta2 | Misuse of research data is a serious problem. | |
| | ta3 | I know the possibility of misuse of research data increases if I do not consider data governance policies and procedures. | |
| | ta4 | I can become a victim of an attack if I do not follow research data governance policies and procedures. | |
| | ta5 | My research data security will be weak if I do not consider data governance policies and procedures. | |
| | ta6 | I do not share harmful research data with the public to reduce risk. | |

**Table A3.** *Cont.*

| Variable | Code | Description of Indicator | Adopted From |
|---|---|---|---|
| Coping Appraisal | ca1 | Currently, the steps I take to prevent unwanted parties from accessing my research data are sufficient. | [51,53,54] |
| | ca2 | Currently, the steps I take to prevent misuse of research data are sufficient. | |
| | ca3 | I have the necessary skills to protect my research data. | |
| | ca4 | I have the expertise to secure my research data from being accessed by unwanted parties. | |
| | ca5 | I believe that the protection of my research data is within my control. | |
| | ca6 | I have the ability to prevent the misuse of my research data. | |
| Data Governance Behavior | dgb1 | I follow data governance policies and procedures to protect research data and prevent misuse of that data. | [55,59] |
| | dgb2 | I consider expert recommendations in conducting research data governance. | |
| | dgb3 | Before taking any action regarding research data governance, I think about the consequences first. | |
| | dgb4 | I consider my previous experience in research data governance to avoid repeating previous mistakes. | |
| | dgb5 | I am always trying to change my habits of practicing research data governance. | |
| | dgb6 | After getting the research data, I have planned from the beginning how the data will be managed, stored, and processed. | |

## Notes

1. https://data.go.id (accessed on 21 September 2022).
2. https://data.brin.go.id (accessed on 21 September 2022).
3. https://peraturan.bpk.go.id/Home/Details/108813/perpres-no-39-tahun-2019 (accessed on 21 September 2022).
4. https://perpus.menpan.go.id/opac/detail-opac?id=2447 (accessed on 21 September 2022).
5. https://jdih.kemenkeu.go.id/in/dokumen/peraturan/cd275d59-a40d-4b48-f105-08d8ac6015bf (accessed on 21 September 2022).
6. https://jdih.brin.go.id/peraturan/view/3b3e5791-f61a-4884-b809-94f1d242ef47 (accessed on 21 September 2022).
7. https://survei.risnov.id (accessed on 21 September 2022).
8. https://sinta.kemdikbud.go.id/ (accessed on 21 September 2022).
9. https://service.elsevier.com/app/answers/detail/a_id/12007/supporthub/scopus (accessed on 21 September 2022).

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
