# Peer review of "Indonesian Scientists’ Behavior Relative to Research Data Governance in Preventing WMD-Applicable Technology Transfer"

_publications, doi:10.3390/publications10040050_

Round 1
Reviewer 1 Report
Topic of the article.
The subject of the article is interesting, relevant and up-to-date.
However, current research, especially in physical and life science, is often international. I have doubts about a survey on data governance conducted in one particular country. The study did not mention whether the respondents take part in international research or whether the standards of openness of research data in other countries influence their behavior. Unfortunately, the survey did not take this issue into account, but the paper should at least address this problem.
Research
Good statistical methods were used in the analysis and the sample was representative.
On the other hand, the response rate was around 15%. The authors wrote that it was good enough. OK, it is high level. However, who was possibly interested in the answer? Perhaps they were scientists interested in the problem of data openness. We do not know this, but it may change the results significantly. This is a common problem with this type of research. It is especially important when the respondents feel they are being evaluated to some extent. Thus, the article is an interesting voice in the discussion, but the detailed number results of the study are not reliable - even when their statistical significance has been proven.
Conclusion
The article, although it has significant problems, may be published after revision. However, the authors should refer to the above remarks in the paper.
Author Response
We thank you very much for the reviewer's valuable comments and suggestions on our manuscript. Our point-to-point response to the reviewer is enclosed below.
Point 1: However, current research, especially in physical and life science, is often international. I have doubts about a survey on data governance conducted in one particular country. The study did not mention whether the respondents take part in international research or whether the standards of openness of research data in other countries influence their behavior. Unfortunately, the survey did not take this issue into account, but the paper should at least address this problem.
Response 1: We thank you for the reviewer's suggestions. We have addressed the comment in subsection 5.3. Limitations and Future Works, as below:
"Many other indicators might influence the pattern of data governance behavior for academics because the factor analysis carried out in this research is based on confirmatory factor analyses. For example, there may be differences in the behavioral patterns of data governance in academia due to the experience of conducting international collaborative research. Different nations have different policies for managing data, what type of data may be shared, and with whom the data are shared."
Point 2: On the other hand, the response rate was around 15%. The authors wrote that it was good enough. OK, it is high level. However, who was possibly interested in the answer? Perhaps they were scientists interested in the problem of data openness. We do not know this, but it may change the results significantly. This is a common problem with this type of research. It is especially important when the respondents feel they are being evaluated to some extent. Thus, the article is an interesting voice in the discussion, but the detailed number results of the study are not reliable - even when their statistical significance has been proven.
Response 2: We thank you for the reviewer's comments. However, we think a 15% response rate, which amounted to 1118 respondents, is sufficient regardless of the participants' interests. We have added an explanation of this claim in subsection 5.3. Limitations and Future Works, as below:
"By using an a priori approach, the required sample sizes for the SEM are 133 and 264 samples for the model's structure and detecting effects, respectively, given the number of latent variables at 9, the number of indicator items at 49, the anticipated effect size of 0.3 (medium), the statistical power level of 0.95, and a probability level of 0.05. Meanwhile, considering the population, the minimum sample size is 348, according to Slovin's formula. Therefore, the response rate of 14.88%, which amounted to 1118 respondents, was considered a sufficient number. However, it would be preferable if the response rate was higher since it would mean more research institutions were catered to in the study."
We also think that even though participants are interested in the problem of data openness, they still need to govern their research data. Governing data does not mean necessarily closing the data but giving access to those eligible to open the data. If scientists need to open the data publicly, they must ensure that the research data are safe and secure, not WMD-applicable. Research data governance rules these measures. Nevertheless, we admitted that it does not rule out the possibility that the participants' interests, such as data openness, could affect the analysis results. Therefore, we have addressed the reviewers' comments at the end of the added paragraph, as below:
"Participants' interests, such as data openness, could also influence the analysis. Future work might explore how data governance practices may vary depending on individual scientists' collaboration patterns, research field, and other factors that this study was not designed to assess."
Reviewer 2 Report
This paper presents the findings of the behavior and attitude of scientist in the specified domain. The paper is well organized and lead the readers from the background through methodology and management implication.
However, the variables, and also the questions in questionaire, mainly focus on human's attitude viewpoint. This paper will be perfect if the below points are added or discussed:
1) The existing industrial or academic data governance frameworks
2) The discussion and conclusions should be more focused on the data management-related functions, e.g. data quality, data security, master data, etc.
3) The data governance organization structure that is needed to advised to the reader.
4) Some of existing regulation that will make the reader see the overall layers: regulation, standards, theory.
Author Response
We thank you very much for the reviewer's valuable comments and suggestions on our manuscript. Our point-to-point response to the reviewer is enclosed below.
Point 1: The existing industrial or academic data governance frameworks
Response 1: We thank you for the reviewer's suggestions. However, data governance frameworks are domain-dependent or country-dependent, and we addressed some of the frameworks in subsection 1.2. Literature Review in the original manuscript.
Point 2: The discussion and conclusions should be more focused on the data management-related functions, e.g. data quality, data security, master data, etc.
Response 2: We thank you for the reviewer's suggestions. We have revised the manuscript by adding a paragraph in subsection 5.2. Practice Implications, as below:
"Research data management (RDM) consists of several activities and processes associated with the life cycle of data, including acquisition, storage, security, preservation, retrieval, sharing, and reuse based on ethical considerations, legal issues, and governance frameworks. RDM is common in developed countries but is a relatively new concept in developing countries, including Indonesia, where national data governance policies are still in its early stages. Indonesian law number 11 of 2019 is the highest legal instrument that regulates RDM in Indonesia. However, it only governs the storage regulation where funders, scientists, and institutions must store the research data in an integrated system. Meanwhile, other activities of RDM still need to be governed. Government regulation derivative from the law has yet to be made available to this day, and policies governing RDM are established by the respective institutions."
Point 3: The data governance organization structure that is needed to advised to the reader.
Response 3: We thank you for the reviewer's comments. Unfortunately, we did not come up with a specific data governance organization structure in this study. However, we have addressed the comment by adding two sentences in subsection 5.2. Practice Implications, as below:
"research institutions are encouraged to establish a structured organization of data governance, where some roles perform multi-level approvals to provide a thorough review process. The organization should be tailored according to the institution's needs since these are likely to vary by country and possibly field of research."
Point 4: Some of existing regulation that will make the reader see the overall layers: regulation, standards, theory.
Response 4: We thank you for the reviewer's suggestions. We have revised the manuscript by adding a paragraph in section 1. Introduction, as below:
"The ODI's portal is an implementation of the Indonesian presidential decree number 39 (from 2019), which is related to the government's data policy, and the purpose of the portal is to create quality data that are accessible and shared across organizations. Previously, the issue of research data governance in Indonesia was addressed by the code of ethics for research activities regulated in the Ministry of Research and Education's decree of 25/M/Kp/III/2013 from 2013. Moreover, Indonesian law number 11 (from 2019), with respect to the national science and technology system, requires that any primary data, including the output of research activities, be stored in an integrated information system. Furthermore, the regulation of BRIN number 18 (from 2022) affirms RIN as an integral part of the national science and technology information system. The Indonesian law number 11 of 2019, Indonesian presidential decree number 39 of 2019, and BRIN regulation number 18 of 2022 express the need for standardized metadata, which is domain-dependent on enabling data reusability and interoperability."